# BRAF Mutant Melanoma Adjusts to BRAF/MEK Inhibitors via Dependence on Increased Antioxidant SOD2 and Increased Reactive Oxygen Species Levels

**DOI:** 10.3390/cancers12061661

**Published:** 2020-06-23

**Authors:** Long Yuan, Rosalin Mishra, Hima Patel, Samar Alanazi, Xin Wei, Zhijun Ma, Joan T. Garrett

**Affiliations:** 1James L. Winkle College of Pharmacy, University of Cincinnati, Cincinnati, OH 45267-0514, USA; yuanlg@mail.uc.edu (L.Y.); mishrarn@ucmail.uc.edu (R.M.); patel2h2@mail.uc.edu (H.P.); alanazsa@mail.uc.edu (S.A.); weix5@mail.uc.edu (X.W.); 2Department of Chemistry, University of Cincinnati, Cincinnati, OH 45267-0514, USA; mazj@mail.uc.edu

**Keywords:** BRAF, melanoma, dabrafenib, trametinib, reactive oxygen species (ROS), superoxide dismutase 2 (SOD2)

## Abstract

B-Rapidly Accelerated Fibrosarcoma (BRAF) mutations are found in about 50% of melanoma patients. Treatment with Food and Drug Administration (FDA)-approved BRAF and MAP/ERK kinase (MEK) inhibitors has improved progression free and overall survival of patients with BRAF mutant melanoma. However, all responders develop resistance typically within 1 year of treatment with these inhibitors. Evidence indicates that reactive oxygen species (ROS) levels are elevated after BRAF pathway inhibition treatment. We aim to decipher the role of mitochondrial antioxidant proteins relative to ROS levels and BRAF pathway inhibitor resistance. We observed BRAF mutant melanoma cells treated with the combination of a MEK inhibitor (trametinib) and a BRAF inhibitor (dabrafenib), exhibited elevated ROS levels, both in in vitro and in vivo melanoma models. We next generated trametinib- and dabrafenib-resistant (TDR) cells and found increased ROS levels after acquisition of resistance. An immunofluorescence experiment showed an increase of DNA damage in TDR cell lines. Furthermore, we observed that TDR cells increased superoxide dismutase 2 (SOD2), an antioxidant, at both mRNA and protein levels, with the upregulation of the transcription factor Nuclear Factor (NF)-κB. Knockdown of SOD2 significantly reduced the growth of BRAF pathway inhibitor-resistant cells. In addition, the results indicate that TDR cells can be re-sensitized to BRAF pathway inhibitors by the ROS scavenger, N-Acetyl Cysteine (NAC). Overall, these data indicate that BRAF pathway inhibitor-resistant cells can compensate for elevated ROS via increased expression of the antioxidant SOD2.

## 1. Introduction

Melanoma, which accounts for 4% of all skin cancers, is the deadliest form of skin cancer, which contributes to 80% of all skin cancer related deaths. About 100,350 new melanomas will be diagnosed and about 6850 people are expected to die of melanoma in the year 2020, according to the American Cancer Society statistics. The identification of the B-Rapidly Accelerated Fibrosarcoma (BRAF)-V600E mutation, which accounts for 50% of patients, piloted a new era in the treatment of advanced melanomas [1,2]. The advent of combination therapy using a BRAF inhibitor (dabrafenib, encorafenib, vemurafenib) with a MEK inhibitor (binimetinib, cobimetinib, trametinib) resulted in a longer median survival rate than other treatments. However, most patients still have relatively short-lived responses with resistance, eventually causing their deaths [3].

Overcoming drug resistance has become an essential issue in metastatic melanoma [4]. Evidence shows that BRAF inhibitors induce reactive oxygen species (ROS) levels in melanoma cells through Peroxisome proliferator-activated receptor gamma coactivator 1 (PGC1α)-induced mitochondria biogenesis [5]. ROS, which consists of superoxide, H_2_O_2_, hydroxyl free radicals etc., is generated due to increased oxidative stress related to enhanced metabolism and the transformation of cancer cells [6]. Recent data from our group indicated that increased ROS activity in dabrafenib-resistant (DR) melanoma cells sensitized these cells to A100, a ROS-activated prodrug that causes DNA double strand breaks. Furthermore, these dabrafenib-resistant melanoma cells had upregulation of the mitochondrial enzyme, manganese superoxide dismutase 2 (SOD2), which functions to reduce ROS levels in the mitochondria [7].

SOD2, a member of the Mn superoxide dismutase (MnSOD) family, is an important mitochondria antioxidant enzyme that protects cells from oxidative stress by converting free superoxide ions (O^2−)^ into hydrogen peroxide (H_2_O_2_). Elevated MnSOD2 levels have been shown to reduce tumor growth, in part by suppressing cell proliferation [8]. It is believed that loss of SOD2 expression propagates early progression of metastatic diseases, while evidence shows that SOD2 levels increase in many tumors as progression goes from the early to late stage of the disease progression [9]. Herein, we found that SOD2 levels are upregulated in trametinib- and dabrafenib- resistant (TDR) melanoma cell lines.

Transcription factor NF-κB proteins are of central importance in inflammation and immunity [10,11]. The SOD2 gene promoter is under control of NF-κB, and thus the activation of the MnSOD gene is in part mediated by NF-κB [12,13]. In addition, evidence has shown that SOD2 is the target of NF-κB, with antioxidant activity [14,15,16]. As a key regulator of SOD2, we observed a significant upregulation in NF-κB in TDR cell lines versus parental drug sensitive melanoma cells. Our data indicate that both NF-κB and SOD2 are upregulated in response to BRAF pathway inhibitor resistance in melanoma cells, which correlated to high oxidative stress.

## 2. Results

### 2.1. ROS Levels Are Increased in BRAF Mutant Melanoma in Response to Treatment with BRAF and MEK Inhibitors

The role of BRAF inhibitors in upregulating ROS levels in melanoma cell lines through PGC1α-induced mitochondria biogenesis has already been reported [5]. We extended our investigations to determine ROS levels after treatment with BRAF and MEK inhibitors. We observed a significant increase in ROS levels after treating BRAF mutant melanoma cell lines (WM-115 and WM-983) with BRAF and MEK inhibitors dabrafenib and trametinib (D+T) for 24 h, as measured using the MitoSOX assay (Figure 1A). ROS level alterations after treating with BRAF and MEK inhibitors in A375 and SK-MEL-24 BRAF mutant melanoma cell lines, are shown in Appendix A with similar results. Next, we assessed hydrogen peroxide (H_2_O_2_) production using established reporter gene cell lines (WM-115-Fluc-Puro and WM-983-Fluc-Puro), which stably express the firefly luciferase gene. We used Peroxy Caged Luciferin-2 (PCL2), a H_2_O_2_-responsive boronic acid probe that releases 6-hydroxy-2-cyanobenzothiazole (HCBT) upon reacting with reactive oxygen species (ROS), as well as D-cysteine, which reacts with HCBT and forms firefly luciferin in situ [17]. Photons and bioluminescence can be released and detected upon reaction with firefly luciferase enzymes in our newly established reporter gene cell lines. We observed an increase of the total bioluminescence signal after WM-115-Fluc-Puro and WM-983-Fluc-Puro cell lines were treated with 2.4 μM dabrafenib and 500 nM trametinib for 24 h, suggesting an increase of ROS levels after treatment of BRAF and MEK inhibitors (Figure 1B). In addition, the WM-115-Fluc-Puro reporter gene cell line was injected in nude mice and tumor xenografts were established. Mice were treated with vehicle or dabrafenib (30 mg/kg/day via orogastric gavage) and trametinib (0.6 mg/kg/day via orogastric gavage) for 3 days. The mixture of PCL2 and D-cysteine (30 μmol each) was injected via intra peritoneal (i.p.) as a probe to measure the bioluminescence signal using the IVIS Spectrum in vivo imaging system [12]. We observed an increase in ROS levels after a short-term 3 day treatment of BRAF and MEK inhibitors, indicating that ROS levels were upregulated in an in vivo mouse melanoma model subjected to acute treatment of dabrafenib and trametinib (Figure 1C).

### 2.2. ROS Level Is Upregulated upon Drug Resistance in BRAF Mutant Melanoma Cell Lines

Our previous data indicated that ROS levels are upregulated in response to acute dabrafenib and trametinib treatment in BRAF mutant melanoma. Our lab has shown that dabrafenib-resistant (DR) cell lines have more ROS levels compared to parental cell lines [7]. We were interested by the alterations in ROS levels in response to dual BRAF and MEK inhibitor-resistant melanoma cells. We generated trametinib- and dabrafenib-resistant (TDR) WM115 and WM983 cell lines via a gradual dose escalation of each drug [18]. We checked superoxide and hydrogen peroxide (H_2_O_2_) levels in TDR melanoma cell lines. Superoxide levels were significantly increased in WM-115 TDR and WM-983 TDR cell lines compared to WM-115 DR, WM-983 DR [7] and parental cell lines, as measured using the MitoSOX assay (Figure 2A). Levels of H_2_O_2_ were measured using the DCFDA/H2DCFDA assay kit to measure ROS, which indicated that WM-115 TDR and WM-983 TDR cells had higher H_2_O_2_ compared to respective DR or parental cell lines (Figure 2B). Overall, the data indicated that ROS levels were significantly augmented in response to chronic treatment with BRAF and MEK inhibitors in TDR melanoma cell lines.

### 2.3. BRAF Mutant Melanoma Cells Resistant to BRAF and MEK Inhibition Show Increases in DNA Damage

Given the fact that acute and chronic BRAF and MEK inhibitor treatment results in upregulation of ROS levels, we assessed the alteration in expression of the DNA damage marker, 8-oxo-dG, in TDR cells versus the DR and parental cell lines via immunofluorescence assay [19]. In nuclear and mitochondrial DNA 8-oxo-dG [20] is one of the predominant forms of free radical-induced oxidative lesions and has therefore been widely used as a biomarker for oxidative stress. Our data indicated that WM-115 TDR and WM-983 TDR have higher levels of 8-oxo-dG than WM-115 DR and WM-983 DR and parental cell lines (Figure 3A,B). These data suggest the possible involvement of DNA damage signaling induced in response to high ROS generated via chronic dabrafenib and trametinib treatment in BRAF mutant melanoma cells.

### 2.4. SOD2 and NF-κB Levels Are Increased in BRAF Pathway Inhibitor-Resistant Melanoma Cells 

Oxidative stress and intracellular redox state balance are essential. Evidence also shows that high intracellular ROS levels could cause an increase in antioxidant enzymes for compensation and homeostasis balance [7,21,22,23]. Based on our previous findings that antioxidants, such as SOD2 and peroxiredoxin-1 (PRDX1), are elevated in dabrafenib resistant (DR) melanoma cells identified from a proteomics screen [7], we aimed to assess SOD2 and PRDX1 levels in BRAF and MEK inhibitor-resistant cells. Two-step real time-qPCR was performed to assess the mRNA level of the antioxidants, SOD2 and PRDX1. Our results show that both SOD2 and PRDX1 antioxidant levels were upregulated at the mRNA and protein levels for WM-115 TDR cell lines versus the dabrafenib resistant (DR) and parental cell lines (Figure 4A). We observed increased mRNA and protein levels of SOD2 in WM-983 TDR versus WM-983 DR and WM-983 parental cell lines. However, we observed a reduction in PRDX1 mRNA and protein levels in both WM-983 DR and TDR cells, as compared to WM-983 parental cells (Figure 4B). NF-κB plays an important role in modulating SOD2 levels as a transcription factor [15,24]. Our results show that NF-κB increased in WM-115 and WM-983 TDR cells, indicating the possible involvement of NF-κB in ROS-mediated adaptive response in the TDR cell lines. Previous data indicated the role of p53 in modulating SOD2 [25]. Since TDR cells are shown to have high SOD2 compared to parental cells, we checked the p53 levels in TDR cell lines. There was no consistent pattern in p53 levels amongst the TDR cells versus the parental melanoma cell lines (Figure 4C). Original whole blot images can be found in Appendix A. Densitometry analysis of western blotting can be found in Appendix A.

### 2.5. Knockdown of SOD2 Suppresses the Growth of BRAF Pathway Inhibitor Resistant Melanoma Cells

Previous reports have shown that SOD2 levels appear to surge during metastatic progression in highly aggressive tumor cells [26,27]. Several studies have shown that elevated antioxidant enzyme expression is necessary for tumor progression and survival [28,29]. We investigated whether knockdown of SOD2 has a role in cell growth and proliferation. WM-115 TDR and WM-983 TDR cells were transfected with siRNA, specifically targeting SOD2 (SOD2 siRNA). The specificity of the siRNA was analyzed by western blotting (Figure 5A). The effect of SOD2 knockdown on WM-115 TDR and WM-983 TDR cell proliferation and growth was assessed using 3-4,5-Dimethyl-2-thiazolyl)-2,5-diphenyl-2H-tetrazolium bromide (MTT) and crystal violet assays. We observed a decrease in cell growth for both TDR cell lines versus the control after SOD2 knockdown, suggesting that SOD2 could be a factor in modulating tumor proliferation and survival in trametinib- and dabrafenib-resistant (TDR) melanoma cell lines (Figure 5B,C). The original whole blot can be found in Appendix A.

### 2.6. NAC, a ROS Scavenger, Can Re-Sensitize the BRAF Inhibitor in Melanoma Cells

We next aimed to examine if we could re-sensitize TDR cells to BRAF and MEK inhibitors via treatment with a ROS scavenger. Hence, we assessed the sensitivity of these cells in the presence of the ROS scavenger, N-Acetyl Cysteine (NAC), using crystal violet and MTT growth assays. We found that TDR cells maintain resistance to dabrafenib and trametinib after drugs have been withdrawn for 21 days. We observed a decrease in cell proliferation and growth of WM-115 TDR and WM-983 TDR cell lines when subjected to the combination of dabrafenib (D) and trametinib (T) in the presence of NAC, compared to the D+T group or DMSO controls (Figure 6A,B). Our data indicated the possible involvement of a ROS-mediated signaling pathway in regulating TDR cell proliferation and growth.

## 3. Discussion

Current targeted therapy for the treatment of the metastatic melanoma with BRAF-V600E mutation focuses on the use of combination of BRAF and MEK inhibitors as a treatment strategy. Several studies and clinical trials have demonstrated improved patient outcomes with this combination therapy. However, there are several challenges met with the current standard of care [4]. Hence, the current research is focused on understanding the resistant mechanisms in response to BRAF and MEK inhibitors. Our study is focused on the role of oxidative stress-induced ROS in trametinib- and dabrafenib-resistant (TDR) melanoma cells with BRAF mutation. Our data indicated that melanoma cells in response to acute and chronic exposure of BRAF pathway inhibitors induce ROS in both in vitro and in vivo melanoma models (Figure 1 and Figure 2) [30,31].

We observed that dual resistant (TDR) melanoma cell lines have higher antioxidant levels (SOD2 and PRDX1), accompanied with higher ROS levels compared to counterpart parental and dabrafenib resistant (DR) cells (Figure 4). The involvement of SOD2 in mediating an effect on proliferation in TDR cells was confirmed by siRNA mediated silencing, where knockdown of SOD2 significantly suppressed melanoma cell proliferation and growth (Figure 5). There is evidence that downregulation or knockout of SOD2 would be lethal in mice models [32,33]. SOD2 is a target for the transcription factor NF-κB. The levels of NF-κB are also influenced by the level of ROS, thus affecting the antioxidant SOD2 levels [15,34,35]. NF-κB binding sites are known to be located in the promoter region and within the intronic enhancer element of the SOD2 gene [36,37,38]. In addition, NF-κB is known to participate actively in the regulation of SOD2 expression in tumor cells [39,40,41]. In our study, we observed an upregulation of NF-κB in the nuclear fraction of BRAF pathway inhibitor-resistant cells relative to parental cells. The increased SOD2 could be caused by NF-κB transcriptional regulation in BRAF pathway inhibitor-resistant cells (Figure 4). Further experiments are needed to delineate if SOD2 levels are directly regulated by NF-κB. p53 levels were upregulated in correlation with enhanced NF-κB levels in the WM-115 TDR cell line. However, there were no alterations in p53 levels in the WM-983 TDR cell line compared to the control. While both WM-983 TDR and WM-115 TDR have increased SOD2, notably, WM-115 TDR cells have much higher levels of SOD2 mRNA compared to parental cells, relative to SOD2 mRNA in WM-983 TDR cells (~50 fold increase in WM-115 TDR cells versus ~6 fold increase in WM-983 TDR cells (Figure 4A,B). Thus, it is possible that WM115 TDR cells have a larger increase in SOD2 levels, as enhanced expression of both the transcription factors (NF-κB and p53) in WM-115 TDR cells may have resulted in increased transcriptional activity of SOD2 [24,42,43].

Furthermore, we observed treatment of BRAF pathway inhibitor-resistant cells with the ROS quenching NAC restored sensitivity to BRAF and MEK, whereas NAC alone had no effect on cell growth (Figure 6). Treatment of cancer with antioxidants, such as Vitamin C, has been studied extensively with inconsistent results. Supplementing diets with the antioxidants NAC and vitamin E in mouse models of BRAF and KRAS lung cancers increased tumor progression and reduced survival [44] by reducing ROS, DNA damage, and p53 expression in mouse and human lung tumor cells. We note the differences in our in vitro administration of NAC in the context of acquired drug resistance. There has been a renewed interest using high dose vitamin C potentially administered intravenous, as there is evidence that large doses of vitamin C can kill cultured colon cancer cells with BRAF or KRAS mutations by raising free radical levels [45].

Our study has several limitations. The bulk of results are obtained in only two cell lines, WM115 and WM983. WM983 is derived from a metastatic melanoma, whereas WM115 is a primary melanoma with metastatic potential [46,47]. Furthermore, the majority of data presented are from in vitro studies and are not derived directly from human patients. We acknowledge the limitations of our data. Nevertheless, crucial information about the interplay between antioxidants, ROS and BRAF pathway inhibitor resistance is obtained from this work, which can be bolstered by future work examining the effects of high dose antioxidants in BRAF pathway inhibitor-resistant tumors in vivo.

## 4. Material and Methods

### 4.1. Cell Culture and Inhibitors

Human melanoma cells (WM-115 and WM-983) were obtained from the American Type Culture Collection (ATCC) (Manassas, VA, USA). Dabrafenib resistant melanoma cell lines were generated in a previous protocol in our lab [7]. Dabrafenib and trametinib were obtained from LC Laboratories. WM-115 cells were maintained in Minimum Essential Medium Eagle (MEM) with 10% fetal bovine serum (FBS) and 1% penicillin-streptomycin at 37 °C in 5% CO_2_ in a humidified incubator. WM983 cells were maintained in Dulbecco’s Modified Eagle’s medium (DMEM) with 10% fetal bovine serum and 1% penicillin-streptomycin at 37 °C in 5% CO_2_.

### 4.2. Generation of Trametinib- and Dabrafenib-Resistant (TDR) Cell Lines

Briefly, melanoma cell lines (WM-115, WM-983) were initially treated with 0.01 μM dabrafenib and trametinib subjected to gradual dose escalation of dabrafenib (0–2.4 μM) and trametinib (0–500 nM) over a span of 3–4 months and finally maintained in 2.4 μM dabrafenib and 500 nM of trametinib. Media and dabrafenib were replenished every alternate day. The cell lines were authenticated using short tandem repeat (STR) profiling by ATCC.

### 4.3. Immunoblotting

TDR melanoma cells were maintained in 2.4 μM dabrafenib and 500 nM trametinib and lysed in RIPA buffer containing protease and phosphatase inhibitors. Nuclear extracts were prepared using the NE-PER nuclear and cytoplasmic extraction reagents Kit (Thermo Fisher, Waltham, MA, USA), as per the manufacturer’s guidelines. Immunoblotting was performed using specific primary antibodies for NF-κB, poly ADP ribose polymerase (PARP), P53, PRDX1, SOD2 (Cell Signaling Technology, Danvers, MA, USA), and Actin (Santa Cruz biotechnology, Dallas, TX, USA). Anti-rabbit IgG-horse radish peroxidase (HRP) was used as the secondary antibody. Actin and PARP served as loading controls.

### 4.4. Transient Transfection

WM-115 TDR and WM-983 TDR cells were transfected with siRNAs, specifically targeting SOD2 (SOD2 siRNA; Dharmacon, Lafayette, CO, USA) and control siRNA (control siRNA; Santa Cruz Biotechnology, Santa Cruz, CA, USA).

### 4.5. MTT Cell Proliferation Assay

Both WM-115 and WM-983 TDR cell lines were transfected with Con siRNA and SOD2 siRNA (WM-115 TDR Con siRNA, WM-115 TDR SOD2 SiRNA, WM-983 TDR Con SiRNA, and WM-983 TDR SOD2 SiRNA melanoma cells). Cells were plated into a 96-well plate at a density of 3.0 × 10^4^ cells per well in triplicate and allowed to grow for 72 h. After 72 h, the media containing the drug was replaced with 5 mg/mL MTT (3-(4,5-dimethylthiazol-2-yl)-2,5-diphenyltetrazolium bromide), dissolved in cell line specific media, and incubated for 4 h. After 4 h, the media was aspirated and crystals were dissolved with isopropanol (Molecular grade, Fisher BioReagents, Waltham, MA, USA). The absorbance was read at 570 nm using a microplate reader (SPECTR Amax PLUS Microplate Spectrophotometer Plate Reader, Molecular Devices Corporation, San Jose, CA, USA). Data is represented as the mean of at least two independent experiments ± SEM.

### 4.6. Crystal Violet Cell Proliferation Assay

Both WM-115 and WM-983 TDR (30,000 cells/well) were transfected using Con siRNA and SOD2 siRNA for 48 h and were plated in triplicate in a 6-well plate. Medium and inhibitors were replenished every 2 days. After 4–6 days, cells were stained with crystal violet in 5% methanol after visible difference was noticed. The intensities were measured using the Odyssey infrared System. The values were expressed as the mean of intensities obtained from three independent experiments and bar graphs were generated using graph pad prism 7 (GraphPad Software, Inc., La Jolla, CA, USA).

### 4.7. Measurement of Intracellular ROS

The production of intracellular ROS was measured using the oxidation sensitive MitoSOX assay using flow cytometry. Cellular O_2_^−^ generation was measured using the mitochondria-targeting probe, MitoSOX Red. WM-115 and WM-983 parental cells (4 × 10^5^) were treated in DMSO, 2.4 μM of dabrafenib, and 500 nM trametinib for 24 h. WM-115 DR and WM-983 DR cells were maintained in 2.4 μM of dabrafenib, while WM-115 TDR and WM-983 TDR cells were maintained in 2.4 μM of dabrafenib and 500 nM trametinib, respectively. The cells were resuspended in Hank’s Balanced Salt Solution (HBSS) and incubated with 5 μM of MitoSOX Red for 30 min at 37 °C. The cells were washed, resuspended in HBSS on ice, and centrifuged at 4 °C. The red fluorescence (MitoSOX, Waltham, MA, USA) were measured using the FL-2 channel of the FACS Diva and analyzed using the CellQuest software.

The production of H_2_O_2_ was measured using the oxidation-sensitive Dichloro-dihydrofluorescein diacetate (DCFH-DA). WM-115 and WM-983 (parental, DR, and TDR) cells (2 × 10^4^) were plated and maintained with DMSO, 2.4 μM of dabrafenib, 2.4 μM dabrafenib, and 500 nM trametinib, respectively. The cells were resuspended in DCFH-DA for 30 min and analyzed using Cell Quest software. The ROS levels were measured and represented using a bar graph.

### 4.8. qPCR Analysis of Antioxidant mRNA

Total RNA was extracted from both WM-115 and WM-983 (parental, DR, and TDR) cell lines using the RNA extraction mini kit (Qiagen, 74104, Germantown, MD, USA), according to the manufacturer’s instructions. Two-step real time-qPCR was performed to assess the mRNA level of SOD2 and PRDX1. First strand cDNA was synthesized using the iScriptTM cDNA Synthesis Kit (Biorad, 1708890, Hercules, CA, USA). qPCR was set up using the CFX96 Real-Time System (Biorad, USA). Actin was used as an internal control. Relative SOD2 and PRDX1 mRNA expression was presented by the 2^−ΔΔCT^ method. Paired primer sequences were used for SOD2: 5’-GCTCCGGTTTTGGGGTATCTG-3’ (forward) and 5’-GCGTTGATGTGAGGTTCCAG-3’(reverse); for PRDX1: 5’- CCACGGAGATCATTGCTTTCA-3 (forward) and 5’-AGGTGTATTGACCCATGCTAGAT -3’ (reverse); and for Actin: 5’AAGGAGCCCCACGAGAAAAAT-3’ (forward); 5’- ACCGAACTTGCATTGATTCCAG-3’.

### 4.9. Immunofluorescence

An equal number of WM-983 and WM-115 (parental, DR, and TDR) were plated on coverslips at a density of 0.15 × 10^5^ cells in each chamber for 24 h. The cells were fixed in 4% formalin and then 0.5% triton for 10 min, respectively. Cells were incubated with 20% bovine serum albumin (BSA) in phosphate buffered saline (PBS) for 1 h at room temperature and analyzed for 8-oxo-dG expression using immunofluorescence assay with antigen retrieval, as described below. Cells (parental, DR, and TDR) were incubated with an antibody mix of 8-oxo-dG mouse monoclonal antibody (Trevigen, 1:1000 dilution) for 2 h at room temperature in a humidified chamber. A total of 5 drops of Red Vectafluor (Vector Lab, Burlingame, CA, USA) fluorescent secondary antibodies were applied to slides for 30 min. Cells were counterstained with a 10 ng/mL DAPI (SigmaAldrich, St. Louis, MO, USA) for 5 min. Immunofluorescence images were captured using a Zeiss fluorescence microscope. Quantification was performed using ImageJ, with three images per treatment group.

### 4.10. Generation of Stable Reporter Cell Lines

Lentiviral vector LV-Luc2-P2A-Puro was purchased from Imanis Life Sciences (Rochester, MN, USA). WM-115 and WM-983 parental cell lines were seeded in complete medium to achieve 60–70% confluency and cells were transduced at a multiplicity of infection (MOI) of 10 in the presence of 6 μg/mL polybrene, as per the manufacturer’s guidelines. Cells transduced with lentiviral vector containing a puromycin (Puro) resistance gene were selected using 625 nM puromycin to generate WM-115-Fluc-Puro and WM-983-Fluc-Puro stable cell lines.

### 4.11. In Vitro Luciferase Assays

A Xenogen IVIS Spectrum instrument (Vontz Core Imaging Laboratory, Cincinnati, OH, USA) was used for bioluminescent imaging in cellular experiments. WM-115-Fluc-Puro and WM-983-Fluc-Puro cells were plated at a density (10^4^, 10^5^, 10^6^ cells/well) of black 96-well plates with clear bottoms and were allowed to grow overnight (Becton Dickinson and Co., Franklin Lakes, NJ, USA). A mixture of PCL2 and D-cysteine (25 mM each) was added and bioluminescence was measured by a Xenogen IVIS Spectrum instrument.

In separate experiments, cells (WM-115-Fluc-Puro and WM-983-Fluc-Puro) were plated in 96-well plates with clear bottoms at a density of 2.0 × 10^5^ cells per well in triplicate and allowed to grow overnight. After 24 h, cells were treated with 2.4 μM of dabrafenib and 500 nM trametinib with DMSO as the control. After 24 h, a mixture of PCL2 and D-cysteine (25 mM each) was added and bioluminescence was measured by a microplate reader (SPECTRAmax PLUS Microplate Spectrophotometer Plate Reader, Molecular Devices Corporation).

### 4.12. In Vivo Xenograft Experiments

Four-week-old, male homozygous Foxn1nu mice were purchased from Jackson Labs and maintained at the University of Cincinnati Laboratory Animal Medical Services (LAMS) facility, with free access to food and water. All studies were approved and performed according to the guidelines of the Animal Care and Use Committee of the University of Cincinnati. Mice were injected subcutaneously with 5 × 10^6^ WM-115-Fluc-Puro cells via the right flank with matrigel (Corning, NY, USA) in a ratio of 1:1. Once tumors had reached a volume ≥200 mm^3^, mice were randomized to the following groups: vehicle control group or 30 mg/kg/day dabrafenib + 0.6 mg/kg/day trametinib oral gavage group (*n* = 3 per group). Control mice were dosed with the vehicle by oral gavage daily for 3 days. Dabrafenib and trametinib were treated daily for 3 days to mimic the standard of care treatment in patients. Dabrafenib and trametinib were suspended in 0.5% methyl cellulose (Sigma, St. Louis, MO, USA) and 1% Tween 80 and administered by orogastric gavage. Tumor diameters were measured serially with calipers and volume in mm^3^.

### 4.13. In Vivo Luciferase Experiments

A Xenogen IVIS Spectrum instrument (Vontz Core Imaging Laboratory) was used for bioluminescent imaging in animal experiments. Mice were anesthetized prior to injection and during imaging via inhalation of isoflurane. Phosphate buffered saline (PBS) was purchased from Thermo Fisher Scientific (Waltham, MA, USA), Isoflurane was purchased from Phoenix Pharmaceuticals, Inc. (St. Joseph, MO, USA), and pharmaceutical-grade DMSO was purchased from Sigma–Aldrich (St. Louis, MO, USA). Mice were imaged via i.p. injection with a mixture of PCL2 and D-cysteine (i.p., 30 μmol each, in 50 μL of 1:1 DMSO:PBS), both for the DMSO and D + T treated groups after a daily oral gavage of D + T. Imaging data were quantified by averaging the region of interest from the maximal photon emitting exposure.

### 4.14. Statistical Analysis

Data are shown as the mean ± standard error of mean (SEM) and are representative of at least three independent experiments, unless indicated otherwise. Statistical analysis was performed by two sample *t*-tests, using ANOVA (Graph Pad Prism 7). The data was considered statistically significant if *p* < 0.05.

## 5. Conclusions

Our study suggests that SOD2 is induced in BRAF mutant melanomas rendered resistant to dabrafenib and trametinib treatment. In addition, SOD2 plays a substantial role in WM-115 TDR and WM-983 TDR cell growth and proliferation. BRAF pathway inhibitor-resistant cells (both WM-115 TDR and WM-983 TDR) had increased DNA damage and increased amounts of the transcription factor NF-κB. Furthermore, BRAF pathway inhibitor-resistant cells can be rendered sensitive to BRAF and MEK inhibitors with the addition of the antioxidant NAC, which reduces ROS levels. This could be a potential strategy to treat BRAF-mutant melanoma patients whose tumors no longer respond to BRAF pathway inhibition.

## Figures and Tables

**Figure 1 cancers-12-01661-f001:**
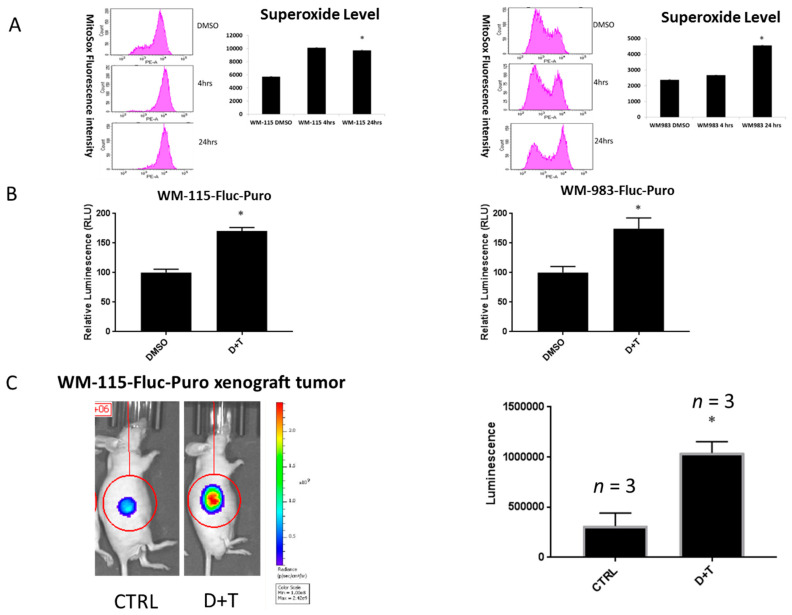
Reactive oxygen species (ROS) level is upregulated upon BRAF pathway inhibitor treatment. (**A**) WM-115 and WM-983 cell lines were treated with 2.4 μM dabrafenib and 500 nM trametinib. Superoxide levels were measured by MitoSOX assay at 0, 4, and 24 h. (**B**) WM-115-Fluc-Puro and WM-983-Fluc-Puro cell lines were treated with 2.4 μM dabrafenib and 500 nM trametinib for 24 h. The total bioluminescent signal, integrated over 5 min, added with a mixture of Peroxy Caged Luciferin-2 (PCL2) (25 mM) and D-cysteine (25 mM) were measured. Signals were normalized to the Dimethyl Sulfoxide (DMSO) control group. (**C**) Left panel: Representative image (45 min post-injection) for mice injected with a mixture of PCL2 and D-cysteine (i.p., 30 μmol each, in 50 μL of 1:1 DMSO: phosphate buffered saline (PBS)) after daily oral gavage of vehicle (CTRL) or dabrafenib and trametinib (D+T) (30 mg/kg/day dabrafenib + 0.6 mg/kg/day trametinib oral gavage for 3 days). Right panel: Total bioluminescent signal quantification (*n* = 3 per group). * *p* < 0.05 versus control group. Statistical analysis by one way ANOVA.

**Figure 2 cancers-12-01661-f002:**
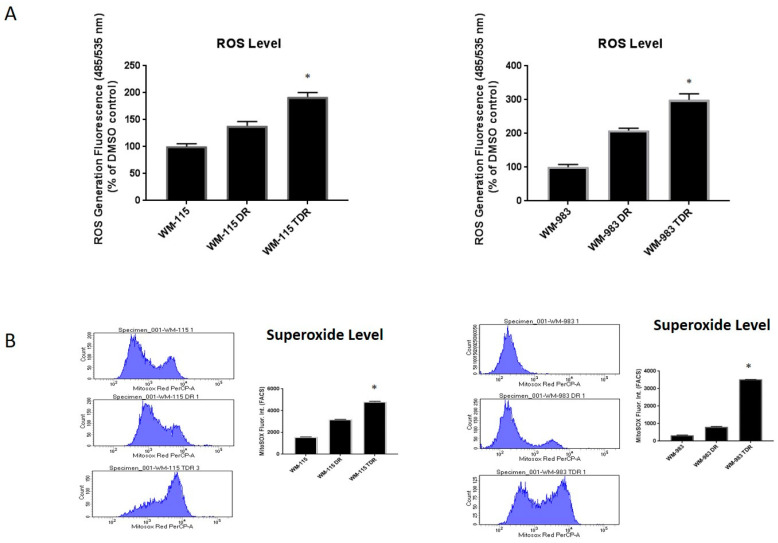
ROS level is upregulated upon BRAF and MEK inhibitor resistance. (**A**) ROS levels in WM-115 (parental, dabrafenib-resistant (DR), and trametinib- and dabrafenib-resistant (TDR) and WM-983 (parental, DR, and TDR) were measured by the DCFDA assay. (**B**) Basal superoxide levels in WM-115 (parental, DR, and TDR) and WM-983 (parental, DR, and TDR) cells were measured by the MitoSOX assay. * *p* < 0.05 versus parental cell lines. Statistical analysis by one way ANOVA.

**Figure 3 cancers-12-01661-f003:**
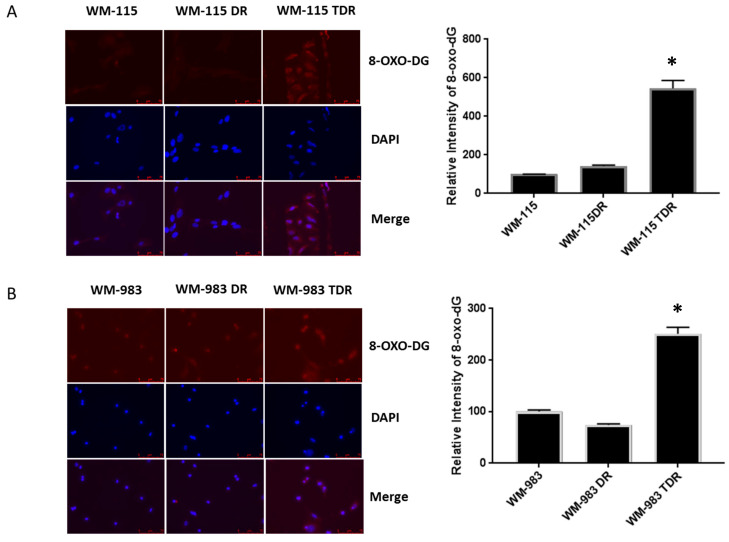
8-oxodG, a DNA damage marker, is increased in BRAF pathway inhibitor-resistant melanoma cells. (**A**) Representative images showing anti-8-oxodG expression for WM-115 (parental, DR, and TDR) cell lines, as detected using immunofluorescent assay. 4′,6-diamidino-2-phenylindole (DAPI) stain is used to identify nuclei. Quantifications are shown on the right panel. (**B**) Representative images showing anti-8-oxodG staining for WM-983 (parental, DR, and TDR) cell lines, as detected by immunofluorescent staining. DAPI stain is used to identify nuclei. Quantifications are shown on the right panel. * *p* < 0.05 versus parental cell lines. Statistical analysis by one way ANOVA.

**Figure 4 cancers-12-01661-f004:**
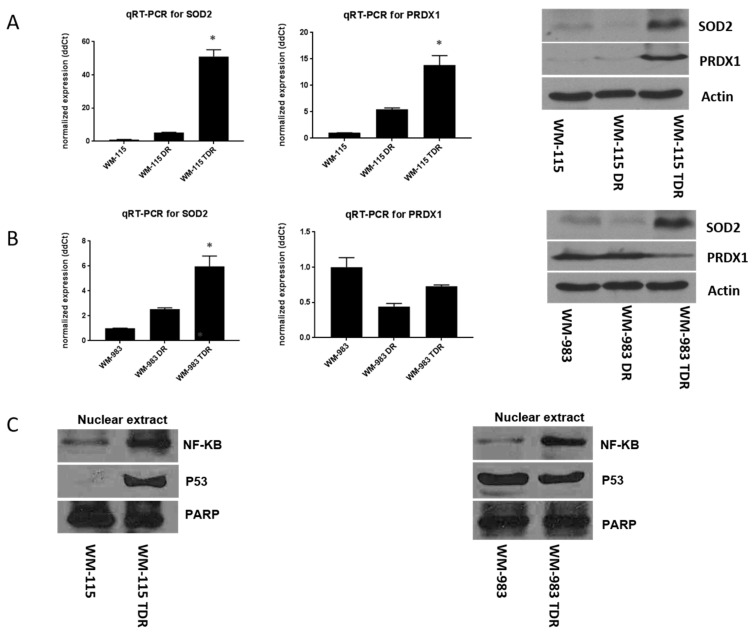
Superoxide dismutase 2 (SOD2) and NF-κB levels are increased in BRAF pathway inhibitor-resistant cells. (**A**) An equal number of WM-115 (parental, DR, and TDR) cells were plated. Total RNA was extracted from these cell lines using the RNA extraction mini kit, according to the manufacturer’s instructions. Two-step real time-qPCR was performed to assess the mRNA level of SOD2 and PRDX1. First strand cDNA was synthesized using the iScriptTM cDNA synthesis kit qPCR, which was set up using the CFX96 real-time system. Actin was used as an internal control. Relative SOD2 and PRDX1 expression was presented by the 2^−ΔΔCT^ method (left panel). In separate experiments, whole cell lysates were prepared and separated in a 12.5% sodium dodecyl sulfate (SDS) gel, followed by immunoblot analysis with indicated antibodies. Western blot data is shown on the right panel. (**B**) An equal number of WM-983, WM-983 DR, and WM-983 TDR cells were plated. Two-step real time-qPCR and immunoblot analysis with indicated antibodies were performed under the above conditions (left and right panels). (**C**) Nuclear extracts were collected and prepared from WM-115 and WM-983 (parental and TDR). * *p* < 0.05 versus parental cell lines. Statistical analysis by one way ANOVA.

**Figure 5 cancers-12-01661-f005:**
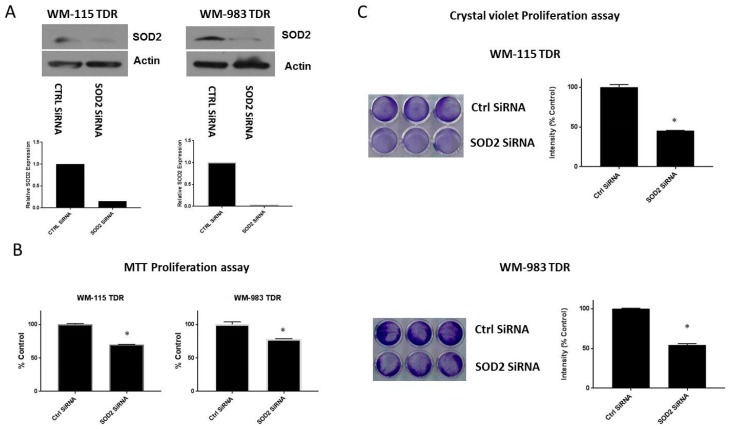
Knockdown of SOD2 reduces growth in BRAF pathway inhibitor-resistant cells. (**A**) WM-115 TDR and WM-983 TDR cells were transfected with a SOD2 specific siRNA or control (siCon) for 48 h and analyzed using specific antibodies. Actin served as the loading control. Densitometric analysis of SOD2 expression was normalized to Actin. (**B**) Both WM-115 TDR and WM-983 TDR cells transfected with Con siRNA and SOD2 siRNA cells were plated in triplicate in 96 well plates for 72 h. (**C**) Both WM-115 and WM-983 TDR cells transfected with Con siRNA and SOD2 siRNA (30,000 cells/well) were plated in triplicate, as indicated. * *p* < 0.05 vs. CTRLsiRNA. Statistical analysis by one way ANOVA.

**Figure 6 cancers-12-01661-f006:**
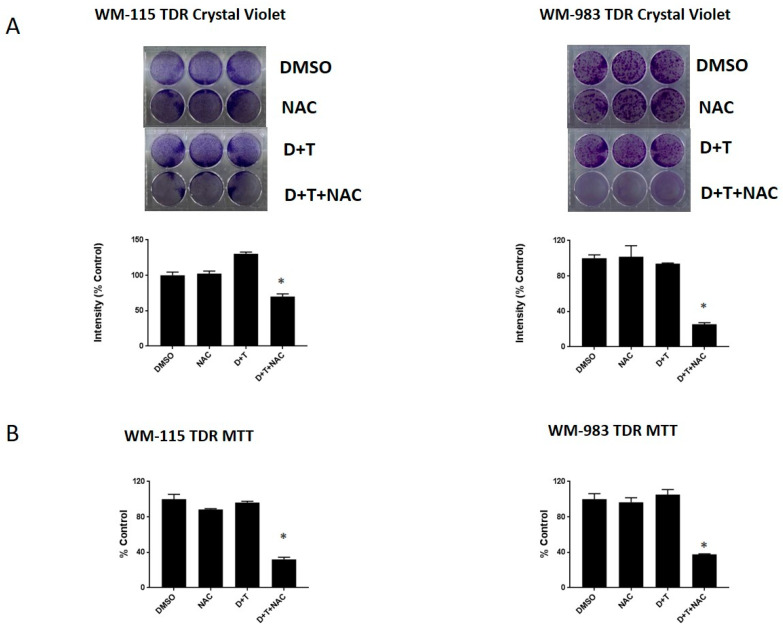
N-Acetyl Cysteine (NAC), a ROS scavenger, can re-sensitize BRAF pathway inhibitor-resistant melanoma cells. (**A**) WM-115 TDR and WM-983 TDR cells (30,000 cells/well) were plated in triplicate and treated with DMSO, 5 mM N-Acetyl Cysteine (NAC), and/or 2.4 μM dabrafenib and 500 nM trametinib, as indicated. (**B**) WM-115 TDR and WM-983 TDR cells were plated in triplicate in 96 well plates and treated for 72 h with DMSO, 5 mM NAC, and/or 2.4 μM dabrafenib and 500 nM trametinib, as indicated. * *p* < 0.05 versus DMSO control. Statistical analysis by one way ANOVA.

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
