# Peer review of "BRAF Mutant Melanoma Adjusts to BRAF/MEK Inhibitors via Dependence on Increased Antioxidant SOD2 and Increased Reactive Oxygen Species Levels"

_cancers, 2020, doi:10.3390/cancers12061661_

Round 1

Reviewer 1 Report

In this manuscript Yuan et al investigated the role of mitochondrial antioxidant proteins relative to ROS level and BRAF pathway inhibitor resistance. This an extension of their published studies that reported that ROS targeted therapy can prolong the efficacy of BRAF inhibitors. The primary conclusion of this study is that BRAF pathway inhibitor resistant cells can compensate for elevated ROS via increased expression of the antioxidant SOD2. Overall, it is a straightforward study with a limited set of experiments on two cell lines- a primary and a metastatic melanoma cell line. However, this manuscript describes a collection of loosely connected experiments that are not fully explored and do not fully support the conclusions.

Specific comments:

  1. Although WM115 is useful as a BRAF-mutant cell lines, since it is a primary melanoma cell line and not relevant as a model for targeted therapy-resistance. Additional metastatic melanoma lines must be used to confirm the data with WM-983 cell line.
  1. The assay with Fluc-Puro cell lines needs clarification. Why was D-cysteine included? A reference must be cited. According to Genevieve C et al (PNAS, 107:21316, 2010) PCL-1 is sufficient to produce the bioluminescent response.
  1. In the Methods it is stated that the TDR cell lines are maintained in dabrafenib and trametinib. If that is the case, data in Fig 6 needs explanation. Are cells treated with DMSO or NAC cultured in medium without the MAPK inhibitors and if so, how long? The effects of NAC must be independently verified using other antioxidants.
  1. What is the status of MAPK activity in TDR cells? Western blots for pERK/ERK must be shown for cells treated with DMSO, NAC and T+D+NAC.
  1. Data in Fig 3 needs quantitation and/or DNA damage must be confirmed with an independent assay.

Author Response

Please see the attached PDF with a response to Reviewer 1.

Reviewer 2 Report

The authors developed two melanoma cell lines resistant to dabrafenib trametinib and they showed that these cell line presented high ROS and DNA damage levels.

They claim that SOD2 upregulation driven by NF-kB can protect resistant cells from high levels of ROS and the use of NAC can restore sensitivity to MAPKi.

Major points:

1) The authors already show similar results in a previous publication using BRAFi resistant cell lines. It has been already extensively demonstrated that BRAFi+MEKi resistant tumors share the same genomic and transcriptomic mechanisms of BRAFi resistance, although augmented. This remarkably reduce the novelty of the findings.

2) The quality of the some Western Blots is quite low (Fig4C and Fig5a) as well as the crystal violet proliferation assay scans (there is also not consistency in the clonogenic growth of the same cell line in different figures)

3) There is no prove in the paper that NF-kB and/or P53 can regulate SOD2 level. the author can try some gain/loss-of-function experiment.

Minor Point

The figure legends are a mix of legend and methods, are quite difficult to follow. It will be better to move the methodological description to the method section.

Author Response

Please see the attached PDF with a response to Reviewer 2.

Reviewer 3 Report

Overall, the method of the experiment and the logic seemed to be appropriate. Authors showed the SOD2 level increased in the D+T resistant cell lines and it could be recovered by NAC. However, authors only showed the results of in vitro experiments. 

Authors need to show the NAC administration (or with vitamin C?) in mice tumor model can recover D+T resistance.  

Is there any method to show the patient with D+T resistance have increased SOD2 level in their tumors?

Author Response

Please see the attached PDF with a response to Reviewer 3.

Round 2

Reviewer 1 Report

The authors made a good effort to address the concerns raised and provided satisfactory responses.

Reviewer 2 Report

The Authors addresses all experimental points

Reviewer 3 Report

Authors added the limitation instead of doing additional experiments. But we should consider the current situation, and therefore, I decided not to ask for the further experiment.